# Socioeconomic Inequalities in Women’s Undernutrition: Evidence from Nationally Representative Cross-Sectional Bangladesh Demographic and Health Survey 2017–2018

**DOI:** 10.3390/ijerph19084698

**Published:** 2022-04-13

**Authors:** Mahfuzur Rahman, Md. Tariqujjaman, Md. Rayhanul Islam, Sifat Parveen Sheikh, Nadia Sultana, Tahmeed Ahmed, Sayem Ahmed, Haribondhu Sarma

**Affiliations:** 1Nutrition and Clinical Services Division, icddr,b, Dhaka 1212, Bangladesh; mahfuzur.rahman@icddrb.org (M.R.); nadia.sultana@icddrb.org (N.S.); tahmeed@icddrb.org (T.A.); 2Health Systems and Population Studies Division, icddr,b, Dhaka 1212, Bangladesh; rayhanul.islam@icddrb.org (M.R.I.); sifatp@yahoo.com (S.P.S.); 3Centre for Injury Prevention and Research, Dhaka 1212, Bangladesh; 4Health Economics and Health Technology Assessment, Institute of Health and Wellbeing, University of Glasgow, Glasgow G1 28RZ, UK; sayem.ahmed@glasgow.ac.uk; 5Department of Tropical Disease Biology, Liverpool School of Tropical Medicine, Liverpool L3 5QA, UK; 6The National Centre for Epidemiology and Population Health, Australian National University, Canberra, ACT 2601, Australia; haribondhu.sarma@anu.edu.au

**Keywords:** undernutrition, ever-married women, inequalities, adolescent, Bangladesh

## Abstract

The objective of this study was to explore the socioeconomic inequalities in undernutrition among ever-married women of reproductive age. We used nationally representative cross-sectional data from the Bangladesh Demographic and Health Survey, 2017–2018. Undernutrition was defined as a body mass index (BMI) of <18.5 kg/m^2^. The concentration index (C) was used to measure the socioeconomic inequality in the prevalence of women’s undernutrition. A multiple binary logistic regression model was carried out to find out the factors associated with women’s undernutrition. The prevalence of undernutrition among women of 15–49 years was 12%. Among them, 8.5% of women were from urban and 12.7% of women were from rural areas. The prevalence of undernutrition was highest (21.9%) among women who belonged to the adolescent age group (15–19 years). The C showed that undernutrition was more prevalent among the socioeconomically worst-off (poorest) group in Bangladesh (C = −0.26). An adjusted multiple logistic regression model indicated that women less than 19 years of age had higher odds (adjusted odds ratio, AOR: 2.81; 95% confidence interval, CI: 2.23, 3.55) of being undernourished. Women from the poorest wealth quintile (AOR: 3.93, 95% CI: 3.21, 4.81) had higher odds of being undernourished. On the other hand, women who had completed secondary or higher education (AOR: 0.55; 95% CI: 0.49, 0.61), married women who were living with their husbands (AOR: 0.72, 95% CI: 0.61, 0.86), and women exposed to mass media (AOR: 0.87, 95% CI: 0.79, 0.97) were less likely to be undernourished. Intervention strategies should be developed targeting the poorest to combat undernutrition in women of reproductive age in Bangladesh.

## 1. Introduction

According to the World Health Organization (WHO), undernutrition is one of the top ten risk factors for the global disease burden [1]. It is the result of the inadequate intake of food in terms of quantity or quality, the poor utilization of nutrients due to infections or other illnesses, or a combination of both factors [2]. Undernutrition and micronutrient deficiencies are prevalent among half of the mothers and children in the world [3], disproportionately affecting the population residing in low- and middle-income countries (LMICs). Maternal undernutrition is most predominant in South Asia, where the prevalence ranges from 10 to 40% [3]. Undernutrition in women at reproductive age leads to adverse outcomes among their off-spring including low birth weight [3], higher risk of anemia, impairment of cognitive and motor development, low productivity in the workplace [4], and increased risk of non-communicable diseases in later life [5,6,7]. During the first half of the ‘first 1000 days of a child’s life’ (from conception to 6 months of a baby’s life), the mother is the entire source of nutrition for the infant via the placenta and then through exclusive breastfeeding during the first 6 months of life, which is recommended by the WHO [8]. 

In LMICs, mothers’ undernutrition accounts for around half of under-five children’s deaths [4]. A pooled estimate of 137 developing countries showed that maternal undernutrition was attributable to 14.4% of stunting among 44.1 million under-two children [9]. On the other hand, women with improved nutritional status are better cared for and provide higher-quality care to their children [10]. Therefore, if the undernutrition among women remains unaddressed, it may have adverse consequences for future generations. To break this vicious cycle of intergenerational undernutrition, we need to understand whether undernutrition among women is overwhelmingly prevalent in any specific group of women in terms of their place of residence, socioeconomic status, and age. The concentration index (C) quantifies the degree of socioeconomic inequality in health and nutrition. Using the C, the degree of socioeconomic inequality can be decomposed into the relative contributing factors, which can provide valuable insights into women’s undernutrition [11].

Bangladesh ranked at position 75 out of the 107 countries included in the Global Hunger Index [12]. About 25% of the population suffered from food insecurity in 2019 in Bangladesh [13], despite the country has made significant progress in achieving food self-sufficiency through agricultural improvement and food production, as well as reducing under-five mortality [14]. Over 15 million people still live in extreme poverty, and their daily earnings are less than USD 1.90, though Bangladesh has achieved sustainable macroeconomic growth [15]. In Bangladesh, around one-third of women of reproductive age are suffering from chronic undernutrition [16]. However, due to the COVID-19 pandemic, as with other LMICs, the circumstances of women’s undernutrition in Bangladesh may be worsening due to increased food prices coupled with the disruption of normal livelihood [17]. 

Previous studies have estimated the prevalence of undernutrition and its associated risk factors among women of reproductive age in Bangladesh [18,19]. However, to the best of our knowledge, no study has provided us with a recent estimation of the prevalence of undernutrition among ever-married women of reproductive age in Bangladesh. Few studies in Bangladesh have analyzed the levels of socioeconomic inequalities in nutritional status using the C, and the majority of existing studies looked into undernutrition among children. There is a dearth of evidence regarding C-based socioeconomic inequalities estimate in women’s undernutrition in Bangladesh and the factors associated with it. Understanding the socioeconomic inequalities associated with undernutrition can guide targeted policy adoption and equitable resource allocation. Therefore, the current study aims to estimate the prevalence of undernutrition among ever-married women of reproductive age in Bangladesh, socioeconomic inequalities in women’s undernutrition, and the factors associated with it. We anticipate that understanding the socioeconomic inequalities of women’s undernutrition will provide us with evidence to design target-oriented interventions for combating undernutrition in women of reproductive age in Bangladesh, and thus help to break the vicious cycle of intergenerational malnutrition.

## 2. Materials and Methods

### 2.1. Data Source

We used data from the nationally representative Bangladesh Demographic and Health Survey (BDHS) 2017–2018, the eighth demographic and health survey which was a part of the global Demographic and Health Surveys (DHS) program. The BDHS 2017–2018 employed a two-stage stratified sampling technique to survey the respondents’ households. In the first stage of sampling, 675 clusters (enumeration areas, EAs) were selected throughout Bangladesh with 250 in urban and 425 in rural areas, with probability proportional to EA size. In the second stage of sampling, systematic sampling of an average of 30 households per EA was selected for urban and rural areas separately, and in each of the eight divisions. The respondents of this survey were ever-married women of reproductive age (15–49 years) (Figure 1). The term “ever-married woman” refers to a woman who has been married at least once in her lifetime. The BDHS 2017–2018 compiles information on a variety of sociodemographic and health-related indicators including the socioeconomic status of the household, fertility and reproductive health, maternal and newborn health, women’s empowerment, healthcare-seeking behavior, knowledge, attitude, and behavior regarding HIV/AIDS and other sexually transmitted infections (STIs), and the nutritional status of women and children.

### 2.2. Outcome Measure 

The outcome variable for this study was the nutritional status of ever-married women of reproductive age. We assessed the nutritional status of ever-married women of reproductive age based on their body mass index (BMI). BMI is defined as weight in kg divided by height in meter square (kg/m^2^). According to the WHO, women with a BMI of <18.5 kg/m^2^ are considered underweight [20]. In this study, we defined the ‘undernutrition’ of women as having a BMI of <18.5 kg/m^2^.

### 2.3. Covariates Measure

To select the covariates relevant to the nutritional status of ever-married women, an extensive literature review was carried out [18,19,21,22,23,24,25,26]. In this study, the covariates were as follows: household size was categorized into <5 members and ≥5 members; respondents’ ages were categorized as 15–19 years, 20–24 years, 25–29 years, 30–34 years, 35–39 years, 40–44 years, and 45–49 years; respondents’ marital statuses were categorized as “others (widowed/divorced/separated)” and “married”; the number of living children was categorized as “no child”, “one child”, “two children”, and “three or more children”; the type of place of residence was categorized as “urban” and “rural”; respondents’ current employment statuses were categorized as “unemployed” and “employed”, respondents’ education levels were categorized as “no formal education”, “primary”, and “secondary or higher”, and administrative divisions (Barisal, Chattogram, Dhaka, Khulna, Mymensingh, Rajshahi, Rangpur, and Sylhet) and wealth quintile (poorest, poorer, middle, richer, and richest) were also used. The DHS constructed the household wealth quintiles based on the household characteristics and ownership of assets using principal component analysis [27], and we kept the same variables and categories for our analysis. Women’s exposure to mass media was characterized in terms of reading newspapers, listening to the radio, or watching television. Responding “yes” to the above-mentioned response options in the survey indicated that the woman was exposed to mass media at least once a week.

### 2.4. Statistical Analyses

All analyses were performed using the statistical software package Stata, version 15.0 (Stata Statistical Software, College Station, TX, USA). Statistical analysis included descriptive statistics of distributions of the study population and the nutritional status of women and presented these in percentages with respective frequencies and 95% confidence intervals (CIs). Bivariate analysis was performed to see the differentials of nutritional status by the selected sample characteristics. In the bivariate analysis, the chi-square test of independence was used to find out the statistical association between nutritional status and sample characteristics. 

We estimated the C to measure the magnitude of the inequalities in the prevalence of undernutrition by asset-based socioeconomic status. The C is the most common measure of socioeconomic inequalities. The C is based on a cumulative frequency curve called the Lorenz curve, which plots the cumulative proportion of outcome (such as women’s undernutrition) against the cumulative proportions of inequality variable (such as the wealth index) [28]. The value of the C usually ranges from −1 to +1; a positive value implies that the prevalence is more concentrated among better-off individuals (such as the poorest group), and a negative value implies the prevalence is more concentrated among the less affluent population (such as the richest group), and the value of 0 indicates no socioeconomic inequalities [29]. Furthermore, the higher the value in either scale (positive or negative), the higher the socioeconomic inequality. 

We also performed simple logistic regression models and presented the results in the crude odds ratio (COR) with 95% CIs to find out the significant associated factors for the multiple logistic regression model. We entered the variables in the multiple model, which were significant at a 5% significance level (*p* < 0.05) in the simple model. Finally, we performed multiple logistic regression to explore the factors associated with women’s undernutrition. We present the results of the multiple regression model in the adjusted odds ratio (AOR) with 95% CI. The prevalence estimation was carried out by taking complex survey design into account for capturing variations due to the weighting and survey design. The variations in the errors due to clustering were also controlled while performing regression analyses.

## 3. Results

### 3.1. Sample Characteristics

This study included 19,798 ever-married women of reproductive age in Bangladesh. Out of them, 10.2% were between 15 and 19 years old, and 16.6% had undertaken no formal education. About one-half of the women were employed, two-thirds of the women were exposed to mass media, and 36% of the women had three or more living children. Around 72% of women lived in rural areas. The distribution of household status in terms of wealth quintile was almost in a similar pattern—around 20% from each category (Table 1). 

### 3.2. Prevalence of Undernutrition 

The prevalence of underweight among ever-married women of reproductive age was 12%. About 9% of these women were from urban areas, and 13% were from rural areas (Figure 2a). We found a significant difference in the prevalence of underweight among women in urban and rural areas by different age categories. The prevalence of undernutrition was 7.4% in urban areas and 12.7% in rural areas among the women who belonged to the 45–49-year age group (Figure 2b). We also found differences in the prevalence of underweight among women by their place of residence in different administrative divisions; it was highest in the Sylhet division (20.8% in total, 17.2% in urban vs. 21.6% in rural) and lowest in the Chattogram division (7.5% in total, 7.4% in urban vs. 7.6% in rural) (Appendix A).

### 3.3. Prevalence and Association of Undernutrition by Sample Characteristics

The prevalence of underweight was higher (21.9%) among adolescents (15–19 years) compared to women of other age groups. Among women with no formal education, the prevalence of underweight was about 15%, and it was 9.2% among women who were exposed to mass media. The prevalence of underweight was 19.8% among women living in the poorest families and 4.3% among women living in the richest families. In bivariate analysis, we found a significant association between a respondent’s age, education, marital status, number of living children, mass media exposure, type of place of residence, wealth status, and undernutrition (Appendix A).

### 3.4. Socioeconomic Inequalities of Undernutrition

We found that the prevalence of undernutrition among women of reproductive age was disproportionately distributed among worse-off socioeconomic groups (C= −0.26; 95% CI −0.28, −0.24). The absolute difference in the distribution of undernutrition was 14.4%, between the poorest and the richest group. Moreover, we found a 4.4 poor (quintile 1): rich (quintile 5) ratio for the prevalence of undernutrition among women of reproductive age in Bangladesh (Figure 3). 

### 3.5. Factors Associated with Women’s Undernutrition

The factors associated with women’s undernutrition are presented in Table 2. In the regression model, after adjusting potential confounders, we found that women of 15–19 years of age and 20–24 years of age had higher odds (AOR 2.81; 95% CI: 2.23, 3.55 for 15–19 years and AOR 1.73; 95% CI: 1.41, 2.13 for 20–24 years) of being undernourished compared to women of 45–49 years of age. Women who completed secondary or higher education and women who were living with their husbands had 45% (AOR 0.55; 95% CI: 0.49, 0.61) and 28% (AOR 0.72, 95%CI: 0.61, 0.86), respectively lower likelihood of being undernourished compared to women with no education and women who were widowed, divorced, or separated. Conversely, women from the poorest and poorer families were 3.93 (AOR 3.93; 95% CI: 3.21, 4.81) and 3.27 (AOR 3.27; 95% CI: 2.69, 3.97) times more likely to be undernourished, respectively, compared to women from the richest families. 

## 4. Discussion

The findings of this study are based on data from a nationally representative cross-sectional survey and reveal that there are socioeconomic inequalities in the prevalence of undernutrition among ever-married women of reproductive age in Bangladesh. Undernutrition was more prevalent among women in the adolescent age group, those residing in rural areas, women in the Sylhet division, and among the socioeconomically worst-off group, based on the C. 

Like other LMICs, socioeconomic inequalities exist [30] in the prevalence of undernutrition among ever-married women of reproductive age in Bangladesh. Our study shows that undernutrition is overwhelmingly prevalent among women from worse-off families, which corresponds to a previous study conducted in Bangladesh [31]. Similar findings were reported in Cambodia, where women in the poorest households were more likely to be undernourished compared to affluent households [32]. The study also reveals that type of place of residence and some demographic characteristics of women such as age are also associated with undernutrition [32]. Additionally, this study demonstrates that marital status, education, and the number of living children of women are also associated with their nutritional status. However, since undernutrition is predominantly prevalent among women from worse-off families, women’s empowerment through multisectoral programs such as conditional and unconditional cash transfer, agricultural intervention, and microfinance can be effective measures for women’s empowerment and can lead to improving the nutritional status of women [33]. The poorest people generally cannot afford to buy nutritious foods, have limited access to healthcare services due to high out-of-pocket expenditure, and have a lack of knowledge about dietary intake. Apart from women’s empowerment, some other modifiable factors need to be addressed to improve the nutritional status of women. 

Nutritional education has been found to be instrumental in improving nutritional knowledge and nutritional behavior among women in LMICs [34]. This study found education to be one of the modifiable factors associated with women’s nutrition. Therefore, the inclusion of nutritional education in the academic curriculum or community-based nutritional education could be an effective way to reduce undernutrition among women. Moreover, nutritional education should be provided during the adolescent period, as the study findings imply that the prevalence of undernutrition is higher among adolescent ever-married women compared to other age groups. The prevalence of undernutrition among married female adolescents is common in other LMICs as well; for example, in Tanzania, 11% of ever-married women of reproductive age suffer from undernutrition, and among them, 18% are from the adolescent age group, which corresponds with our findings [26]. The higher prevalence of undernutrition among married female adolescents is alarming because adolescents are yet to gain up to 50% of their final adult weight and 15% of their final adult height [35]. If growth is halted during the adolescent period, it will have adverse consequences in adulthood, and thus, it is very likely to have an effect on pregnancy outcomes [36].

This study has some noteworthy strengths. The use of large, nationally representative survey data allowed us to examine the regional variations in undernutrition among women across the country. Moreover, the study employed a standard parameter, the C, to examine the significance of socioeconomic inequality on women’s undernutrition. There were a few limitations. Since cross-sectional data were used, it was not possible to establish a cause–effect relationship between women’s undernutrition with each factor. We considered low BMI (underweight) as the only measure of undernutrition, while other parameters such as anemia were not included. Moreover, BMI is not stable over time and may demonstrate potential fluctuations. Future studies should include other parameters in addition to underweight and longitudinal data to provide a more comprehensive overview of undernutrition among women in Bangladesh. 

## 5. Conclusions

Undernutrition is highly prevalent among rural and female adolescents. A significant inequality of women’s undernutrition exists between the poorest and richest households. The results of this study imply that multifaceted problems including unemployment, lack of formal education, and not being exposed to mass media are exacerbated by socioeconomic inequalities in women’s undernutrition. The major problems include the higher prevalence of undernutrition among adolescents and women from the poorest families. Thus, we recommend target-oriented and multisectoral intervention strategies including women’s empowerment through education, poverty alleviation programs, and mass media campaigns to address the problems to combat undernutrition in women of reproductive age in Bangladesh and other similar resource-poor settings.

## Figures and Tables

**Figure 1 ijerph-19-04698-f001:**
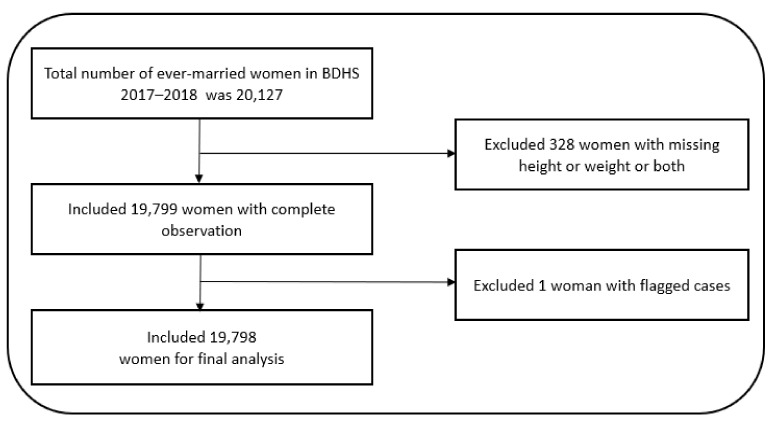
Flow-chart, selection of study participants.

**Figure 2 ijerph-19-04698-f002:**
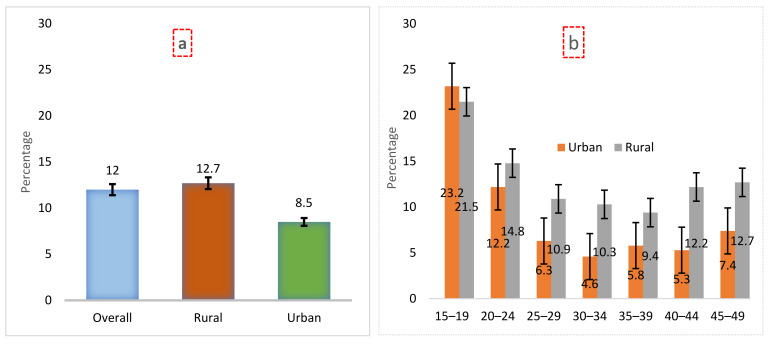
(**a**) Overall and residence-specific prevalence of undernutrition among women of reproductive age by place of residence; (**b**) Age-specific prevalence of women’s undernutrition by urban–rural.

**Figure 3 ijerph-19-04698-f003:**
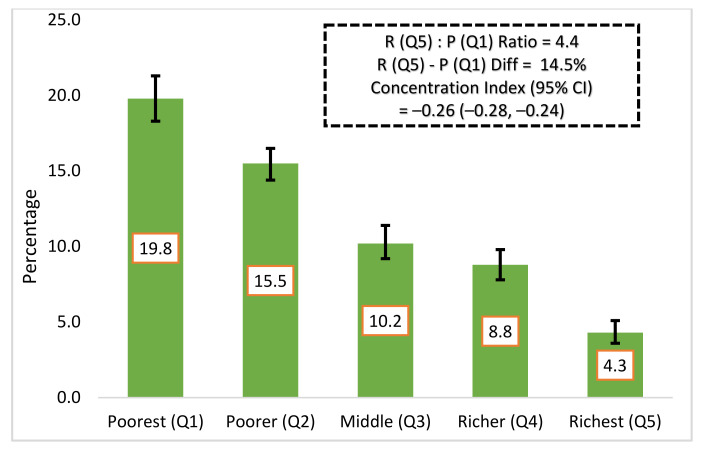
Socioeconomic inequalities in women’s undernutrition in Bangladesh.

**Table 1 ijerph-19-04698-t001:** Sample characteristics of the study population.

Characteristics	Number	Percentage	95% CI
Number of household members			
<5	8648	44.6	43.5, 45.8
≥5	11,150	55.4	54.2, 56.5
Age of respondents			
15–19 years	1916	10.2	9.7, 10.8
20–24 years	3455	17.6	17.0, 18.3
25–29 years	3518	17.8	17.2, 18.4
30–34 years	3410	17.2	16.6, 17.9
35–39 years	2902	14.3	13.8, 14.8
40–44 years	2287	11.4	10.9, 11.9
45–49 years	2310	11.4	10.9, 11.8
Respondents’ education			
No formal education	3202	16.6	15.8, 17.5
Primary	6340	31.4	30.5, 32.4
Secondary or higher	10,585	52.0	50.6, 53.3
Respondents’ marital status			
Others (widowed/divorced/separated)	1232	5.7	5.3, 6.0
Married	18,895	94.4	94.0, 94.7
Respondents’ current employment status			
Unemployed	10,280	52.1	50.2, 53.9
Employed	9518	47.9	46.1, 49.8
Number of living children			
No child	2044	10.5	10.0, 11.0
One child	4559	22.7	22.0, 23.4
Two children	6141	30.8	29.9, 31.7
Three or more children	7054	36.0	35.0, 37.0
Mass media exposure			
No	6888	34.1	32.3, 36.0
Yes	12,910	65.9	64.0, 67.7
Type of place of residence			
Urban	7193	28.1	27.3, 29.0
Rural	12,605	71.9	71.0, 72.7
Wealth quintile			
Poorest	3784	18.7	17.2, 20.3
Poorer	3798	19.8	18.8, 20.9
Middle	3849	20.3	19.3, 21.4
Richer	4037	20.9	19.7, 22.1
Richest	4330	20.3	18.9, 21.8
Administrative division			
Barisal	2126	5.6	5.3, 5.9
Chattogram	2840	17.9	17.2, 18.6
Dhaka	2873	25.1	24.3, 26
Khulna	2599	11.7	11.2, 12.1
Mymensingh	2148	7.8	7.2, 8.3
Rajshahi	2553	14.0	13.5, 14.7
Rangpur	2470	11.9	11.4, 12.5
Sylhet	2189	5.9	5.7, 6.2

**Table 2 ijerph-19-04698-t002:** Factors associated with undernutrition among women of reproductive age.

Variables	COR	95% CI	*p*-Value	AOR	95% CI	*p*-Value
Household size (Ref. <5)						
≥5	1.09	0.99, 1.2	0.071			
Age of the respondents (Ref. 45–49)						
15–19	2.30	1.92, 2.74	<0.001	2.81	2.23, 3.55	<0.001
20–24	1.35	1.16, 1.57	<0.001	1.73	1.41, 2.12	<0.001
25–29	0.91	0.77, 1.08	0.285	1.17	0.96, 1.42	0.115
30–34	0.75	0.63, 0.9	0.002	0.90	0.74, 1.09	0.273
35–39	0.78	0.65, 0.94	0.008	0.85	0.70, 1.02	0.086
40–44	0.95	0.79, 1.15	0.626	0.99	0.82, 1.19	0.885
Education level (Ref. No formal education)						
Primary	0.81	0.72, 0.91	0.001	0.81	0.72, 0.91	0.001
Secondary or higher	0.55	0.49, 0.61	<0.001	0.55	0.49, 0.61	<0.001
Marital status (Ref. Others)						
Married	0.69	0.59, 0.81	<0.001	0.72	0.61, 0.86	<0.001
Employment status (Ref. Unemployed)						
Employed	1.18	1.08, 1.29	<0.001	1.13	1.03, 1.24	0.013
Type of place of residence (Ref. Urban)						
Rural	1.62	1.47, 1.78	<0.001	0.98	0.88, 1.09	0.743
Mass media exposure (Ref. No)						
Yes	0.52	0.48, 0.57	<0.001	0.87	0.79, 0.97	0.012
Number of living children (Ref. No child)						
One	0.79	0.68, 0.91	0.001	0.93	0.80, 1.09	0.356
Two	0.52	0.45, 0.60	<0.001	0.80	0.67, 0.95	0.012
Three or more	0.63	0.55, 0.72	<0.001	0.87	0.72, 1.06	0.168
Wealth quintile (Ref. Richest)						
Poorest	5.44	4.61, 6.41	<0.001	3.93	3.21, 4.81	<0.001
Poorer	4.23	3.58, 5.0	<0.001	3.27	2.69, 3.97	<0.001
Middle	2.65	2.22, 3.16	<0.001	2.30	1.90, 2.78	<0.001
Richer	2.09	1.75, 2.51	<0.001	1.83	1.52, 2.21	<0.001
Division (Ref. Barisal)						
Chattogram	0.67	0.55, 0.81	<0.001	0.85	0.70, 1.05	0.127
Dhaka	0.84	0.69, 1.01	0.064	1.23	1.01, 1.51	0.038
Khulna	0.95	0.78, 1.14	0.572	1.20	0.99, 1.46	0.066
Mymensingh	1.72	1.44, 2.06	<0.001	1.66	1.38, 1.99	<0.001
Rajshahi	1.10	0.92, 1.33	0.292	1.23	1.01, 1.49	0.036
Rangpur	1.28	1.06, 1.53	0.008	1.19	0.98, 1.43	0.075
Sylhet	2.07	1.74, 2.46	<0.001	2.38	1.98, 2.86	<0.001

COR = crude odds ratio, AOR = adjusted odds ratio, CI = confidence interval.

## Data Availability

Data are available in a public, open-access repository. All data related to the study are included in the manuscript.

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
