# Peer review of "Socioeconomic Inequalities in Women’s Undernutrition: Evidence from Nationally Representative Cross-Sectional Bangladesh Demographic and Health Survey 2017–2018"

_ijerph, 2022, doi:10.3390/ijerph19084698_

Round 1

Reviewer 1 Report

The authors accepted the suggestions and revised the manuscript accordingly.

This manuscript is a resubmission of an earlier submission. The following is a list of the peer review reports and author responses from that submission.

Round 1

Reviewer 1 Report

The reviewed manuscript provides insights into problem of socioeconomic inequalities in women undernutrition in Bangladesh. Understanding this problem can help to break the vicious cycle  of intergenerational malnutrition.  

Remarks for Authors: 

      I suggest including the latest publications on female malnutrition in the COVID-19 context, e.g.: Shekar M, Condo J, Pate MA, Nishtar S. Maternal and child undernutrition: progress hinges on supporting women and more implementation research. Lancet 2021, 397(10282),1329-1331. doi: 10.1016/S0140-6736(21)00577-8.

Response: Thanks. We have gone through the paper and elaborated our background section in light of this paper and also cited this paper (page 2, lines: 84-86). This is as follows

“However, due to the Covid-19 pandemic, like other LMICs, the situation of women undernutrition in Bangladesh may be getting worst due to the food prices coupled with disruptions of normal livelihood.”

    It would be worth referring to the socioeconomic situation of Bangladesh.

Response: We have included the socioeconomic situation of Bangladesh in the background section (page 2, lines: 77-83).

“Bangladesh ranked at 75th position out of the 107 countries according to the Global Hunger Index [12]. About 25% of the population remained food insecure in 2019 in Bangladesh [13], despite the country has made significant progress in achieving food self-sufficiency through agricultural improvement, food production, as well as reducing under-five mortality [14]. Over 15 million people still live in extreme poverty and their daily earnings are less than US$1.90, though Bangladesh has achieved sustainable macroeconomic growth [15]

    I miss the policy recommendations part. What comes out from the analysis? In particular, more details on public health interventions that could reduce reproductive age women undernutrition are needed. 

Response: We have included the policy recommendation in the conclusion section as (page10, lines: 311-315)

“Therefore, we recommend target-oriented and multisectoral intervention strategies including women empowerment through education, poverty alleviation programme, and mass-media campaign to address the problems and thus to combat undernutrition in women of reproductive age in Bangladesh and other similar resource-poor settings.”

    The authors should point out the limitations of the study.

Response: We have already pointed out the limitations of the study as (page 9, line: 299-303).

However, the study has a few limitations. Since cross-sectional data was used, it was not possible to establish a cause-effect relationship of women’s undernutrition with the factors. We considered low BMI (underweight) as the only measure of undernutrition, while other parameters such as anaemia were not included. Moreover, BMI is not stable over time and may demonstrate potential fluctuations. 

    The “Conclusions” section should be extended.

Response: According to your suggestion, we have elaborated on the conclusion section as (page 10, lines: 306-315).

“Undernutrition is highly prevalent among rural and adolescent women. A significant inequality of women's undernutrition exists between the poorest and richest households. The results of this study imply that multifaceted problems including unemployment, no formal education, and being unexposed to mass-media are exacerbated by socioeconomic inequalities in women undernutrition. The major problems include the higher prevalence of undernutrition among adolescents and women from the poorest families. Therefore, we recommend target-oriented and multisectoral intervention strategies including women empowerment through education, poverty alleviation programme, and mass-media campaign to address the problems and thus to combat undernutrition in women of reproductive age in Bangladesh and other similar resource-poor settings.”

    The legend in Figure S1 should be corrected  (the supplementary file).

Response: Thanks a lot. Corrected the legend.  

Reviewer 2 Report

The article proposes an analysis of undernutrition among Bangladeshi women through a quantitative study conducted on the basis of the data of the Bangladesh Demographic and Health Survey, 2017-2018. 

Response: Thank you. We used the latest BDHS data in this study.

The study has the merit of highlighting some of the striking trends emerging in the data pool. However, its analysis appears superficial. The article does not provide any economic, social, or historic account of the country and how the pieces of evidence can be read in light of the ongoing literature concerning, for example, rural development, urban poverty, the political and religious culture of the country. In this respect, the contribution to the debate appears limited and its readability. 

Response: Thanks for your valuable comment and insight. Accordingly, we have mentioned in detail for constructing the wealth index. There are some limitations in the dataset because it does contain the variables of social and historical aspects of Bangladesh. However, we used the composite index for constructing the wealth Index.

We applied the concentration index to measure the socioeconomic inequalities which have been applied in many studies.

A robust revision and implementation of the paper are required in order to make it publishable. Specifically, because the method of analysis used does not appear particularly innovative so to make the paper striking at least in terms of methodological advancement. 

Response: Thanks for your valuable comments. As per your other comments. We have made substantial changes in the method section and hope that the revised version of the manuscript will present methodological robustness.

In the endeavor, from Sen onwards, there are several contributions the authors can use. Specifically, I would suggest they have a look to the contribution given by anthropologists to the debate. E.g. Parvanta et al 2007, Karim 2016, Rashid 2007.

Response: As per your suggestion, we have gone through the literature.